# Climate-Related Sea Level Rise and Coastal Wastewater Treatment Infrastructure Futures: Landscape Planning Scenarios for Negotiating Risks and Opportunities in Australian Urban Areas

**Kaihang Zhou** [1] [iD] and **Scott Hawken** [2,*] [iD]

1 Aspect Studios Adelaide, Adelaide 5000, Australia
2 School of Architecture and Civil Engineering, University of Adelaide, Adelaide 5005, Australia
* Correspondence: scott.hawken@adelaide.edu.au

**Abstract:** Around the world, human populations and their supporting infrastructures are concentrated in coastal areas. With rising sea levels, these settlements and urban infrastructures are at risk of service interruptions, lasting damage and frequent climate-related hazards. Wastewater systems are especially vulnerable due to their proximity to coastlines. Despite the seriousness of sea-level-rise-induced challenges, a clear understanding of the risks and potential adaptations of coastal wastewater treatment systems and their associated landscapes in Australia has been overlooked. Further, there is a lack of urgency and awareness concerning this issue. In this study, we consider how scenario-based landscape design approaches might enhance current debates and approaches related to coastal change with particular reference to wastewater treatment systems and associated environmental landscapes. Adelaide is used as a case study, and a range of landscape planning exploratory scenarios are developed and evaluated to assess the possible consequences of different courses of action in uncertain contexts. We find that whilst wastewater treatment plants are threatened by climate-related hazards, there is an opportunity for landscape-scale environmental planning to manage risks and opportunities and improve ecological and economic outcomes. We also find that for wicked multidimensional problems, such as sea level rise, landscape scenario design testing can assist in identifying a number of creative adaptation approaches that are not immediately apparent. We find that approaches such as retreat, defense and accommodation are not mutually exclusive but can each share elements and strategies. The strategic potential of a more creative, scenario-based approach can therefore form a productive part of the sea level rise adaptation of coastal infrastructure landscapes in Australia and elsewhere.

**Keywords:** sea level rise; climate change adaptation; coastal infrastructure; green infrastructure; wastewater treatment; landscape design; environmental planning; landscape planning; geodesign; coastal morphology; coastal squeeze

## 1. Introduction

Sea level rise has occurred throughout the 20th and 21st centuries and will continue to rise due to climate change [1]. The magnitude of future sea level rise is uncertain and depends on differing climate adaptation pathways. However, we do know that the level of rise will be substantial [1]. By 2100, global sea levels could rise by more than one meter when the loss of glaciers is considered [2,3]. In some cases, sea level rise could be even more extreme [4]. In Australia, sea levels have risen at an average rate of 1.6 mm annually over the past four decades, and it is projected to be 0.52–0.98 m by 2100 for the highest greenhouse gas emissions scenario [5]. According to the IPCC's sixth report, the average rate of sea level rise increased about three times between 2006 and 2018 [3].

Worldwide, there are 680 million people living in low-lying coastal zones that are threatened by rising sea levels. With current urbanization trends, three times this population is expected to live and be located in coastal areas by 2050. In total, 50% of the world's population will be within 100 km of coastal areas. As a coastal country, Australia conforms to these trends, with more than 50% of its population living within 7 km of the coast [6]. In the next forty years, another 6.8 million people are expected to inhabit coastal areas due to migration and population growth [7]. Within this group, approximately six per cent of Australian addresses are within three kilometers of the shoreline and in areas that are less than five meters above mean sea levels [6].

To support coastal settlements, extensive infrastructure is also located in coastal zones. Such coastally located infrastructures are particularly susceptible to sea level rise and its associated risks, including coastal flooding, seawater intrusion and storm surges. With future sea level rise, such events will be intensified [1,8]. Due to sea level rise and the related risks and impacts on infrastructure systems, coastal societies face multiple evolving hazards, including the immobilization of transportation, blackouts, saltwater intrusion of water supplies and cascading failures that can reverberate throughout the whole settlement system [9–12]. Insensitive urban development can interact with climate-related risks to exacerbate environmental vulnerability and intensify socio-ecological inequity [13,14].

Wastewater treatment plants (WWTPs) are particularly vulnerable when compared with other coastal infrastructures due to their inherent characteristics. For example, in order to minimize the need for energy and cost, coastal WWTPs are typically located at low elevations to collect the consumed water before discharging it through pipelines to adjacent water bodies under the action of gravity [15]. WWTPs are also particularly centralized [16,17] when compared with other infrastructure, such as transport and power generation systems, which have in-built redundancy. Therefore, WWTPs are more vulnerable due to their centralized structure [18]. Lastly, as WWTPs depend on transport and energy systems to function and be maintained, the failure of such systems can also have knock-on effects, leading treatment systems to fail [18]. If WWTPs services are interrupted or damaged due to sea level rise, extensive populations can be affected well beyond the zones directly inundated [18].

The objective of this paper is to identify the future potential effects of SLR and extreme coastal flooding events on WWTPs and their surrounding landscapes with a focus on the metropolitan area of Adelaide. It aims to provide helpful approaches for safeguarding both natural and urban infrastructure and built wastewater infrastructure in Australia and globally. Further, we aim to investigate and develop possible landscape planning measures to mitigate and adapt to the impacts of SLR by using landscape scenario approaches. These include defensive-, accommodation-, retreat- and business-as-usual-based approaches.

Scenario-based approaches are used to help decision-makers and stakeholders manage the inherent uncertainties of complex situations. Such approaches can involve experimentation and creative decision making, helping to open up possibilities and previously unforeseen opportunities. Further, they can also help diffuse tense and adversarial situations that can arise when stakeholders remain stuck within institutional and disciplinary silos [19,20]. Scenario-based approaches, therefore, generate multiple possibilities and allow the comparison of diverse outcomes in different possible future contexts [21–24]. Such an approach has been widely utilized in urban and landscape planning since the early 1960s [25,26]. Researchers have made significant efforts to improve such approaches and have demonstrated the application of the approach in different fields such as land use policy, environmental systems, economic development and transportation, among others [22,27–36].

In this study, we consider and ask, "What are the climate related threats to Wastewater Treatment Systems and their adjacent landscapes and how can we better generate ways of adapting such systems and their landscapes through landscape planning and design approaches?"

The paper is organized in the following way: we first summarize the current state of knowledge concerning WWTS through a literature review, and following this, we adopt a case study approach to investigate and understand the risks related to wastewater systems and their landscapes in Adelaide, South Australia. Then, a series of landscape scenarios are developed and evaluated for the Bolivar Wastewater Treatment Plant, Adelaide, Australia. The outcomes of this analysis and scenario testing are then discussed in relation to the challenges of WWTS climate-related risks.

## 2. Background

### 2.1. Climate-Related Risks to Wastewater Treatment Systems

There is a range of threats and risks related to coastally located WWTS. The current literature describes these risks and provides an evidence base for the subsequent scenario testing involved in this study. According to such current research, the foremost threat to WWTPs is flooding caused by climate-change-related sea level rise [18]. As sea levels rise, WWTPs situated in coastal areas with low elevation may be subjected to permanent flooding or frequent nuisance flooding due to high tide levels, other extreme weather events, such as storm surges, or a combination of these. In addition to the flooding exposure of coastal WWTPs, increased sea levels can also block outfalls from the system or reduce the efficiency of discharge [18]. Without the assistance of additional or larger pumps and pipelines, the flow rates will cease or decrease, causing siltation and effluent backflow, leading to further maintenance and repair costs [15]. Furthermore, the debris left by the flooding may also block the inlets or outlets of pipelines and cause major damage [15].

In addition to flooding, coastal storms are one of the most serious threats to coastal communities, resulting in huge human and economic losses every year [37–39]. The risk to WTTS due to flooding caused by coastal storms has been recognized as a worldwide problem. For example, in September 2004, Hurricane Ivan produced a 4-m-plus storm surge in Pensacola, USA, which resulted in the local WWTP experiencing four days of significant flooding and power outages [40]. More recently, a storm hit Colorado in 2013. The WWTP was breached, and massive quantities of untreated wastewater polluted the sea [41]. In recent decades, extreme coastal storms and related hazards have intensified due to SLR and show an upward trend. In Australia, about 87% of the total economic damage each year is caused by weather-related factors, mostly due to floods, storms and tropical cyclones [42]. With future climate change and SLR expected to continue, extreme events will continue to intensify and become more frequent. In the next 100 years, Australia will experience increasing coastal vulnerability to its many WWTS (Figure 1) and other infrastructure due to climate-change-related events [43]. As summarized in Table 1, sea level rise poses a number of direct threats and impacts on WWTPs.

**Table 1.** Sea level rise (SLR)-induced effects on WWTPS in the literature.

| SLR Impacts on Wastewater Treatment Plant |
| --- |
| SLR can induce land subsidence and alter topography and pipeline gradients, increasing the risk of sewer overflow and the rupture of WWTP pipelines and utilities [18,44,45]. |
| SLR can increase groundwater levels and subsequent saltwater intrusion into aquifers, leading to the corrosion of sewer pipes and other WWTP infrastructure [46,47]. |
| SLR-related groundwater level increases can exert uplift forces on pipelines and utilities, again causing rupture and altering flows [48]. |
| SLR-related saltwater intrusion can result in increased maintenance costs for wastewater treatment plants due to the need for a desalination process [18]. |
| Excessive water can overwhelm WWTPs, resulting in wastewater backup and flooding in nearby residential or low-lying areas [18]. |
| The weight of floodwaters has the potential to cause structural damage to wastewater treatment plants, resulting in untreated wastewater being discharged into the surrounding environment [49]. |
| Due to groundwater table levels rising WWTTPs may experience areal flooding, causing them to cease functioning [15,18]. |
| Debris from sea-level-rise-related flooding has the potential to cause blockages within pipe inlets, outlets and pipelines [15]. |
| 'Inflow' can occur, involving the entry of water into a sewage collection system through surface apertures following a flooding event, causing WTTPs and related systems to cease functioning [47]. |

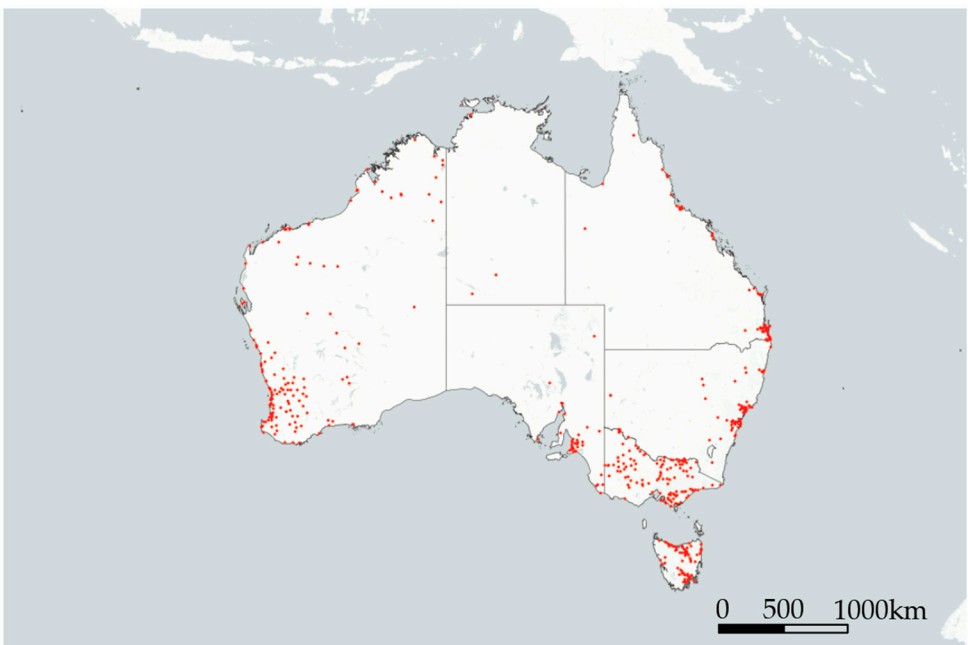

**Figure 1.** Map of the 511 wastewater treatment plants in Australia with their locations shown in red. In total, 321 of them are located along the coast. Many of these are under the threat from sea level rise (Figure by authors using data from Geosciences Australia).

*2.2. Interactions between Wastewater Treatment Systems and Coastal Ecosystems*

Within South Australia, wastewater treatment systems are frequently in close proximity to coastal ecosystems. Such wetlands typically include seagrasses, intertidal mangroves and supratidal samphire or salt marsh vegetation. Coastal ecosystems, especially mangroves, are well known for their ecological importance, providing breeding and nursery sites for crustaceans, shellfish, fish, birds and mammals. In addition, they contribute a range of significant economic and social services as a form of green–blue infrastructure. For example, they help ameliorate greenhouse gas emissions through carbon sequestration. According to Danielsen [50], although mangroves account for only 0.5% of coastal areas worldwide, they contribute 10–15% in coastal sediment carbon storage and export 10–11% of terrestrial particulate carbon to the ocean. They also help to significantly attenuate wave energy to mitigate coastal storm surge hazards. Experimental models have demonstrated that 30 mangrove trees per 100 m$^2$ in a 100 m wide belt have the potential to attenuate the maximum tsunami flow pressure by more than 90% [51]. More recently, Sun and Carson [39] quantified the disaster cost avoidance savings delivered by mangroves and coastal wetlands. However, coastal ecosystems are under threat globally due to climate change and other human-related impacts, such as land use change. Alongi [52] indicates that one-third of mangrove forests have disappeared during the last 50 years. Such environmental degradation processes will exacerbate sea level rise and increase the number of threats related to coastal WWTPs and other infrastructure.

One of the most serious factors associated with sea level rise resulting in coastal wetland system loss has been described as 'coastal squeeze' [53–55]. Even though mangroves and other coastal ecosystems often have the capacity to migrate as sea levels rise, such inland movement is often blocked or "squeezed" by natural or built topographic conditions [55,56]. For example, the inland movement of coastal ecosystems can be blocked by topographic features, such as embankments, roads, seawalls or natural dune systems [56–58]. We, therefore, consider such coastal ecosystems as a critical element in developing and shaping future coastal landscapes.

## 3. Methods and Approach

This study uses exploratory [59–61] landscape design scenario methodologies and is consistent with that used by others in relevant fields, such as conservation [62], landscape architecture [63] and planning [64], as a form of research through design. Exploratory scenario planning aims to expand thinking and consider possible outcomes rather than develop a preferred or better-performing outcome. In the words of Peterson et al. [62], exploratory "scenarios are alternative, dynamic stories that capture key ingredients of our uncertainty about the future of a study system. Scenarios are constructed to provide insight into drivers of change, reveal the implications of current trajectories, and illuminate options for action".

The scenarios were developed within an educational setting in a landscape architecture studio at the University of Adelaide. Such approaches using multiple climate scenarios have been developed to encourage lateral thinking and to generate unforeseen possibilities in uncertain contexts [65,66]. The approach borrows from the geodesign [64,67] process, which uses geospatial analyses to inform landscape planning. Scholars have argued for the use of such approaches to support transdisciplinary thinking and integrate diverse considerations in creative ways [68].

The scenario method involves a series of considered steps informed by data and current literature and research to define the analytic framework and the scenarios themselves. The specific steps that have been used are presented in Table 2.

**Table 2.** Detailed overview of the flooding simulation data sources used in the study.

| Data | Details | Source |
|---|---|---|
| Digital Elevation Model (DEM) | A 1 m × 1 m resolution DEM model from Geoscience Australia 2013 was used as the base for the simulation | See Geoscience Australia [69] |
| Sea Level Rise Estimation by 2100 | SLR under five different GHG emission scenarios from IPCC 2022 report was used for the simulation | See Masson-Delmotte et al [3] and Pörtner et al [70]. |
| Mean High-Water Spring Tide (HAT) | The Port Adelaide HAT at 1.51 m was identified from the Department of planning, transport and infrastructure 2019 Tide Tables | See BOM [71] |
| Storm surge extent | The highest record of storm surge at 1.5 m along Adelaide coast was used in the simulation | See Bourman et al [72]; seeAdelaide's Living Beaches: A Strategy for 2005–2025 [73]. |

### 3.1. Identifying a Focal Issue or Decision

Initially, we orientated our study around climate change and wastewater infrastructure and its relation to a complex mix of land uses, as outlined in the background of this paper. In particular, we focus on coastal ecosystems and related issues such as "coastal squeeze". As mentioned above, the coastal squeeze phenomenon involves the loss of coastal habitats as they are flooded, eroded and hemmed in by sea defenses and other natural or artificial topographic features in the face of climate-change-related sea level rise [36,58]. Although ecosystems and settlements everywhere are being placed under pressure through climate change, the coastal interface is one of the most intense and consequential areas of focus globally.

### 3.2. Identify Driving Forces, Systems, Uncertainties and Focal Landscapes

We then identified climate-change-related sea level rise as the key driving force, along with its interactions with contextual topographic elements and wastewater treatment across the metropolitan area of Adelaide. Sea level rise was tested and identified through a series of simulations within a GIS system informed by IPCC [3,70] projections. The SLR simulation and analysis involves compiling data on the mean sea level rise and data on

extreme events, such as high tide and storm surges, and then assessing the flood risks using this data within the geographic information system (GIS).

In this study, the flooding simulation on site for each of the four scenarios was delineated by using a passive flood model or a so-called "modified bathtub method" in a Geographic Information System (GIS). The passive flood model involves projecting a flood surface onto a Digital Elevation Model (DEM), with the flood surface being a horizontal plane with a predetermined height or elevation, such as sea level rise, which is commonly used to provide SLR adaptation guidance for local planning [74].

Table 2 summarizes the sources and characteristics of the data used in this study.

Using this approach, a focal landscape—the Bolivar Wastewater Plant—was identified for scenario testing.

### 3.3. Select Scenario Logics and Creation of Alternatives

Next, we considered possible ways of managing and working with sea level rise and climate-related pressures as they relate to urban systems, wastewater treatment and the landscape. This involves creative approaches to topographic elements and the development of interventions to construct new ecologies for future landscapes in the face of current driving forces.

### 3.4. Build and Elaborate the Scenarios

Through a process of mapping the infrastructure from open data sources [75,76], we developed an understanding of the various land use systems at play. We used three logics [77] to identify and build the scenarios. These include (i) the prioritization of ecosystem reconstruction through retreat, (ii) the prioritization of urban development and infrastructure through defense and (iii) a mixed hybrid accommodation scenario that involves a combination of both through adaptation.

### 3.5. Evaluate the Scenarios

For each scenario, we evaluated it in terms of ongoing risk and management, which was informed by the literature set out in the introduction and background. The exposure to these was evaluated using a nominal scale and developed in a studio setting. For each scenario, we identified current and future economic, social, and ecological consequences of the course of action and discussed policy implications.

## 4. Results

### 4.1. Case Study Selection

To select case study sites, three criteria were considered, as follows: (a) the wastewater treatment on the site should depend on a centralized WWTP, which is potentially vulnerable to flooding in the future as a result of sea level rise; (b) the coastal landscape should include ecosystems that are potentially threatened by the rising sea levels; and finally, (c) the site should be within Metropolitan Adelaide, South Australia. Adelaide is situated on the Adelaide Plains, within the northern Fleurieu Peninsula. As the capital city of South Australia, the Greater Adelaide region has a population of approximately 1.36 million, which accounts for 77.6% of the state's total population and covers an area of 3259.836 km$^2$. Along with a 90 km coastline of the Gulf St Vincent, Adelaide also contains 13 major catchments and 12 estuaries. The climate in Adelaide is Mediterranean and characterized by hot, dry summers and wet winters, which seasonally flush the rivers and creeks.

Since the settlement of Adelaide in 1836, the treatment of sewerage has always been a critical issue. After years of development and upgrade, the four centralized WWTPs (Bolivar activated sludge, Bolivar High Salinity, Glenelg and Christies Beach) along the coast have become the core of the whole system. The sewage of the city was initially collected by night carts and then disposed of through the use of open channels and cesspits from 1848 [78]. In 1879, a 1.9 km$^2$ sewerage farm was constructed at Islington, which resulted in a 40% decline in deaths in Adelaide from 23.5 per thousand in 1880 to 14.3 per

thousand in 1886 [79]. In 1971, the last major metropolitan facility, Christies Beach WWTP, was constructed to serve the rapidly expanding southern suburbs. There are now four major WWTPs (Figure 2) located near the coastal populations and receiving sewage from more than 455,000 houses and businesses in Adelaide, treating more than 250 mega-liters of wastewater per day through a 7200 km long wastewater network grid.

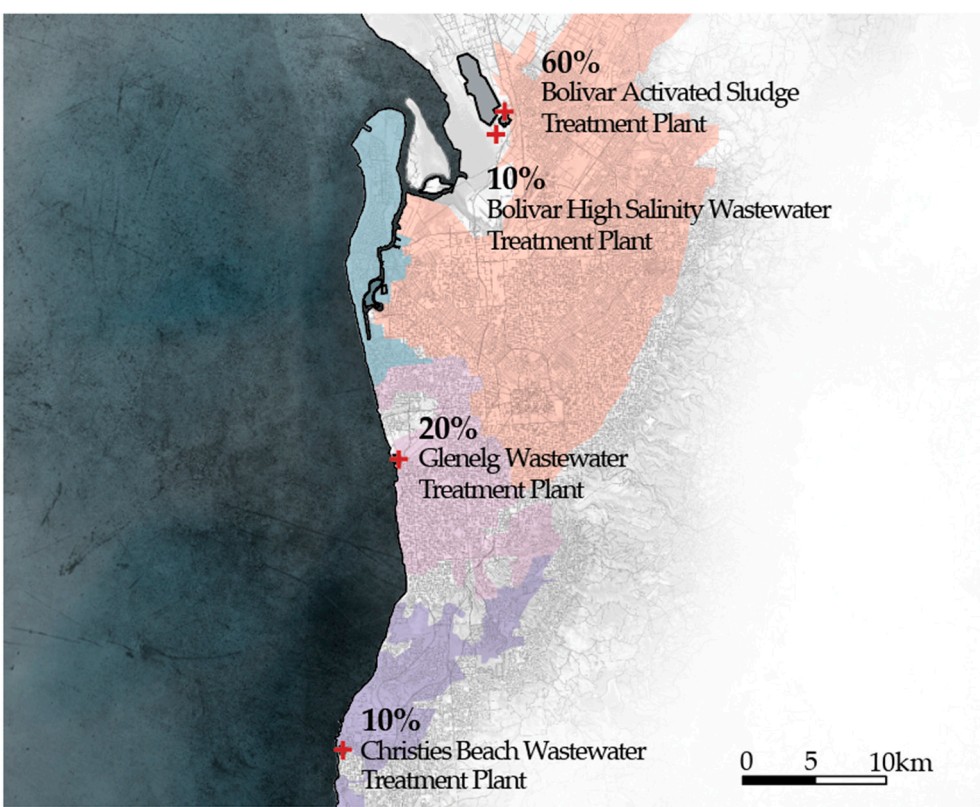

**Figure 2.** Among the four main four WWTPs in Adelaide, the Bolivar Wastewater Treatment Plant, consisting of the Bolivar Activated Sludge Treatment Plant and the Bolivar High Salinity Wastewater Treatment Plant, treats more than 70% of the wastewater per day. (Figure provided by authors using data from Geosciences Australia).

*4.2. Results of Simulation Testing*

To project the sea level rise scenario, the IPCC [3,70] provides future potential sea levels, with global mean sea level projection in the sixth assessment report. A set of five new sea level rise scenarios were completed based on five greenhouse gas (GHG) emission projections: SSP5-8.5 (Very High), SSP3-7.0 (High), SSP2-4.5 (Intermediate), SSP1-2.6 (Low) and SSP1-1.9 (Very Low) [18]. According to the report, by 2021, the sea level rise under the Low (SSP1-2.6) and Very Low (SSP1-1.9) scenarios is expected to reach 0.32–0.62 m and 0.28–0.55 m, respectively, with GHG emissions declining to net zero around 2050. SLR is expected to be 0.44–0.76 m higher under the Intermediate scenario (SSP2-4.5), with GHG emissions maintained around current levels until the middle of the century. The High scenario (SSP3-7.0) and the Very High scenario (SSP5-8.5) are calculated in the context of GHG emissions that roughly double from current levels by 2100, and with SLR reaching 0.55–0.90 m higher and 0.63–1.02 m, respectively. In addition, Mean High-Water Spring Tide (HAT) at 1.51 m, the highest record of storm surge at 1.4 m [72,73], was applied in conjunction with SLR to simulate different flooding conditions. A one-meter accuracy digital model download from the government topographic online database "ELVIS" was used in the analysis.

Sea level rise simulations were completed for the three WTTPs for future sea levels under five different GHG emission scenarios (see Figure 3). The Bolivar Wastewater Treatment Plant is the most vulnerable (see Figure 4). Even under the very low GHG

emission scenario (SSp.1-19) (see Figure 3), during the highest astronomical tide (HAT), seawater levels could reach the levees of the stabilization lagoon of the plant, and the lagoon will likely be flooded if an extreme event occurs. Under the Very High scenario (SSp.5-8.5), the seawater driven during a HAT can reach the plant. As for the Glenelg Wastewater Treatment Plant, the threat mainly comes from the landward Patawalonga River to the east of the site. In contrast, the Christies Wastewater Treatment Plant is relatively unaffected by the sea level rise. Even under the extreme event of the Very High scenario (SSp.5-8.5), the seawater could not reach the plant. Based on these simulations, the Bolivar Wastewater Treatment Plant (BWWTP) in Adelaide was selected as the focus case study site.

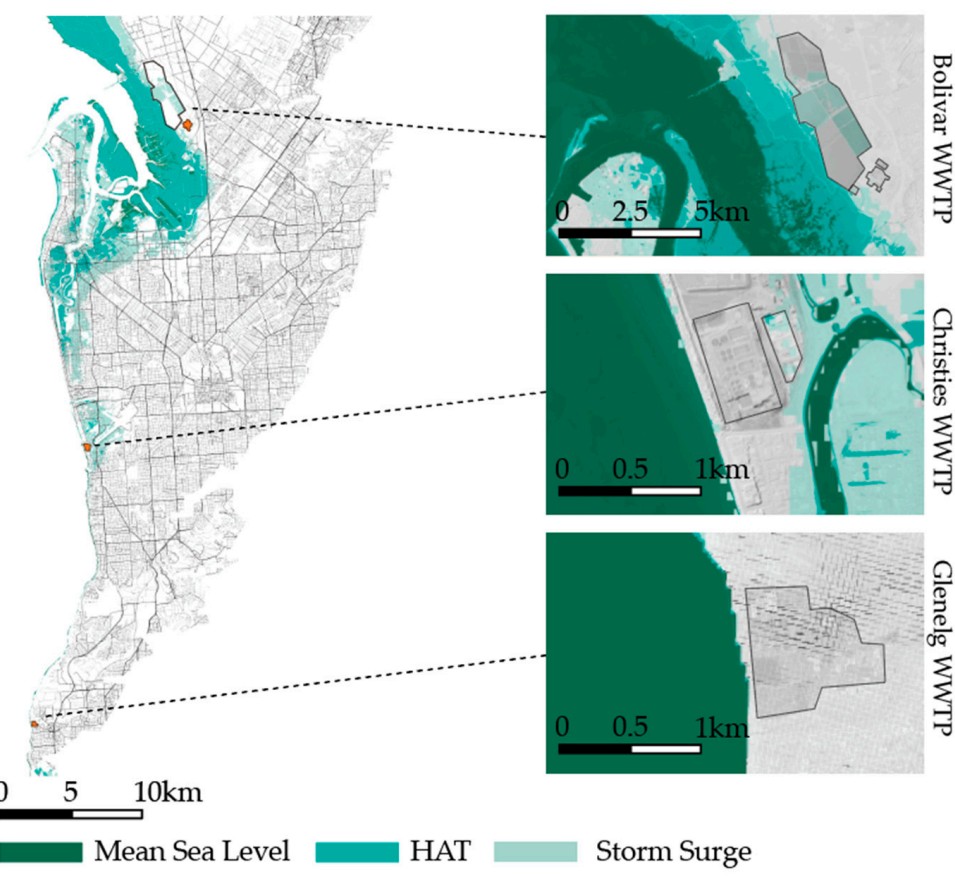

**Figure 3.** The sea level rise simulation of the three WWTPs in Adelaide according to the SSP 1-1.9 IPCC scenario. The grey elements represent terrestrial landscapes and infrastructure, and the green and blue elements represent tidal areas under the particular sea level rise scenario. According to the various simulations, the Bolivar Wastewater Treatment Plant (BWWTP) is the most vulnerable to sea level rise, followed by Glenelg. Christies is relatively unaffected (figure by authors, using data from Geosciences Australia [69] and the IPCC [3,70] see Table 1).

As the largest centralized WWTP among the four in Adelaide, the Bolivar Wastewater Treatment Plant (BWWTP) is located in the Northern Adelaide Region, 20 km northwest of the city center and near the suburb of Elizabeth. BWWTP consists of Bolivar activated Sludge Treatment Plant and Bolivar High Salinity Treatment Plant. Since it was established in 1966, the BWWTP has experienced several upgrades, including the replacement of the former trickling filters with an "activated sludge process" in 2001 and the construction of the Bolivar High WWTP to replace the Port Adelaide WWTP in 2004. Additionally, a 22 million Virginia Pipeline Scheme (VPS) has been constructed, which transfers treated wastewater to the Virginia agricultural area for irrigation. Currently, the integrated plant treats more than 70% of metropolitan Adelaide's wastewater and serves approximately one million

customers from 15 local municipalities (the City of Playford; Salisbury; Port Adelaide Enfield; Tea Tree Gully; Charles Sturt; West Torrens; Prospect; Campbelltown; Walkerville; Norwood; St Peters; Burnside; Adelaide; Unley; and Mitcham) [80]. As it is located less than one kilometer from the coast, with future sea level rise, the Bolivar WWTP is threatened by coastal flooding and other hazards. According to an online interactive mapping tool unveiled by Coastal Risk Australia, under a 0.84 m sea level rise by 2100 scenario, seawater incursion will reach the levees defining the plant's lagoon.

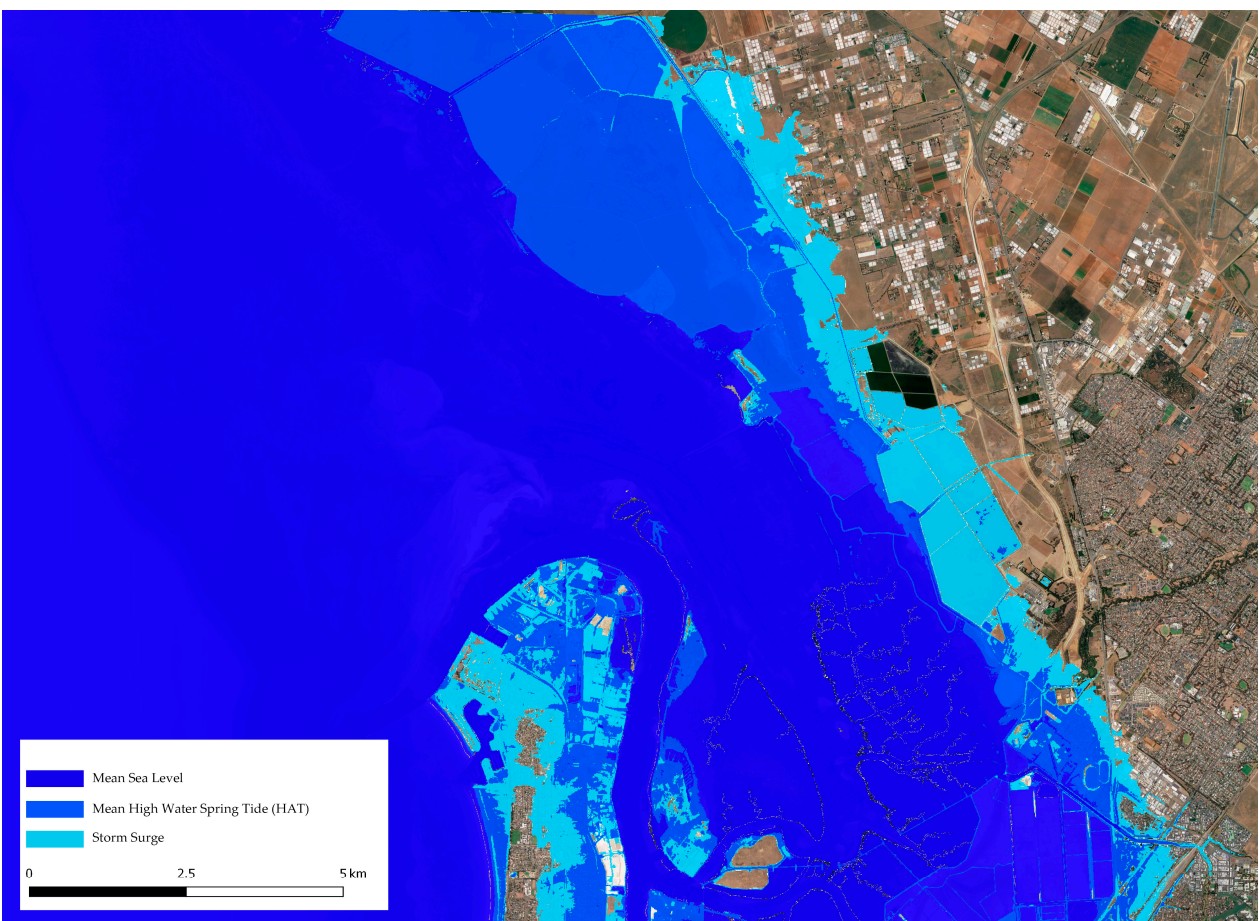

**Figure 4.** The figure shows that the Bolivar Wastewater Treatment Plant is vulnerable to flooding by 2100 under the High Predicted Inundation Scenario. The light blue shading indicates the extent of storm surge, the mid blue shading indicates the high spring tide extent, and the deep blue indicates the mean sea level (Figure by authors using data from Geosciences Australia).

In addition, the coastal area near the WTTP site is well known for its ecological, commercial and recreational value and hosts a range of coastal and estuarine habitats. A band of coastal wetland systems consisting of a seaward fringe of seagrass meadows, extensive intertidal stands of mangroves and supratidal saltmarsh vegetation occupies more than 1000 hectares. Such ecosystems provide an ideal habitat and food resources for a number of marine creatures, including dolphins and migratory shorebirds [81]. Additionally, as one of the most important fish nursery areas in the state, it supports more than 11 important commercial and recreational fish and crustacean species [82], which contribute up to AUD 379 million to the state's economy, and a total of 3108 jobs in the state [83]). However, with increases in sea level rise, a series of former salt-production ponds located behind the WTTP could block inland ecosystem migration, leading to 'coastal squeeze' and the disappearance of valuable wetland systems, as schematically indicated in Figure 5. The salt fields were established in the 1930s and now stretch 28 km along the coastline, occupying approximately 5500 ha of land. Operation largely ceased with the closure of the Penrice

soda ash factory. With the removal of commercial production, currently, some ponds have dried out, while others were reopened to the sea and have become tidal. Environmental pollution, such as effluent discharge and the leakage of hypersaline brine, has sometimes occurred on site, which has led to the degradation of the wetland system. During 1935–2005, the outfall of Bolivar WWTP at the north of St Kilda discharged 35.3 GL/yr of treated effluent to the coastal environment, which led to 900 ha of intertidal and subtidal seagrass and intertidal mangroves being lost [84]. Additionally, a recent leakage event at salt fields near St Kilda was recorded, which led to the death of 100 ha of mangroves and saltmarshes.

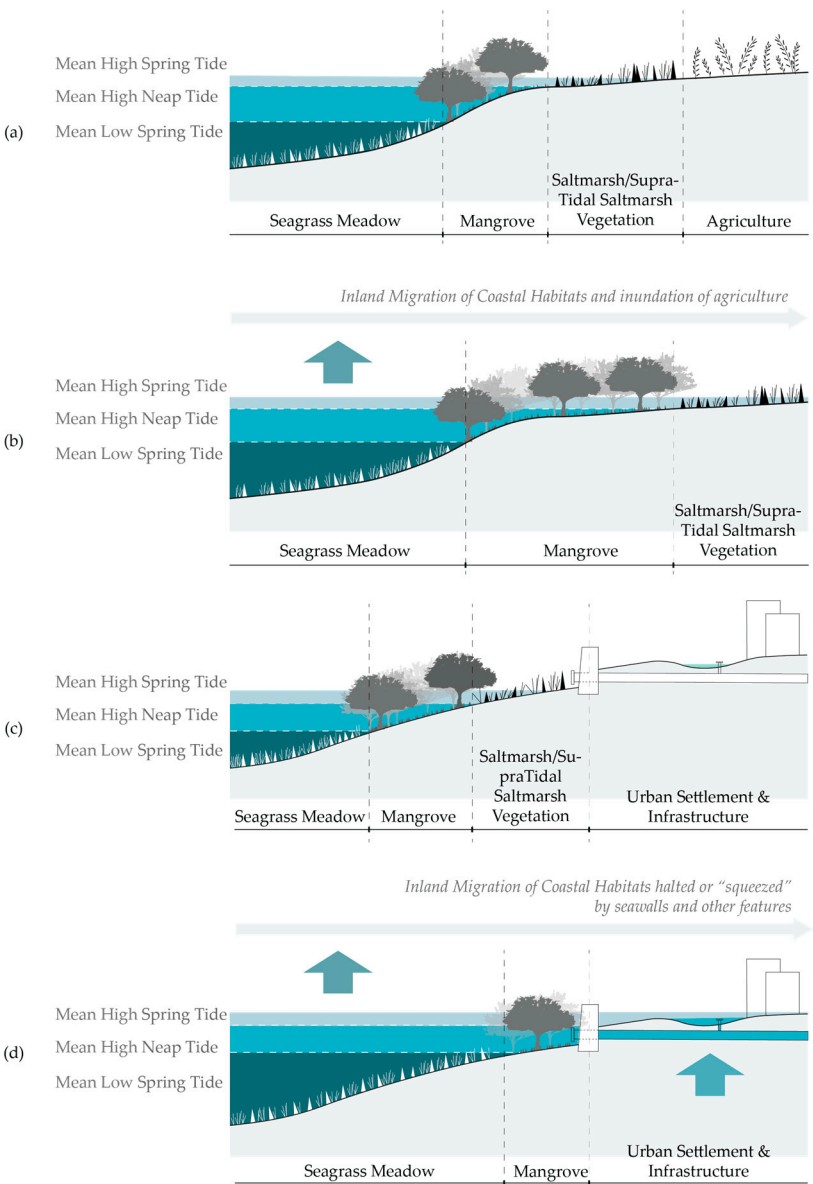

**Figure 5.** The figure shows the coastal squeeze concept for both agricultural–coastal interface and urban–coastal interface. (**a**) shows a natural intertidal gradient under current conditions. (**b**) shows a natural intertidal condition under climate-related sea level rise in which habitats migrate to higher elevations as the water level rises. (**c**) shows the current urban sea–land interface characterized by a mix of land uses and with a sea wall and other infrastructure. (**d**) shows the same urban sea–land interface experiencing coastal squeeze due to the sea wall, which blocks the inland migration of habitats. Area flooding is also occurring behind the seawall due to the water table increasing.

### 4.3. Results and Discussion of Scenario Testing

Following the selection of the Bolivar Wastewater Treatment Plant as a focal landscape, four scenarios were developed using landscape scenario design approaches and contrasting strategies and principles. These include business as usual, inland retreat, defend and accommodation. Each integrated the main topographic features, shown in Figure 6, around the Bolivar WWTP in diverse and creative ways. The four scenarios are set out in Figure 7. Then, the economic and ecological values for each scenario were assessed by using the Land Use Generalized layer, Australian Exposure Information Platform (AEIP) [85], layer in a GIS, as well as introducing the cost data for sea wall construction [86] and the ecological value of coastal ecosystem [87] based on similar infrastructure projects recently implemented.

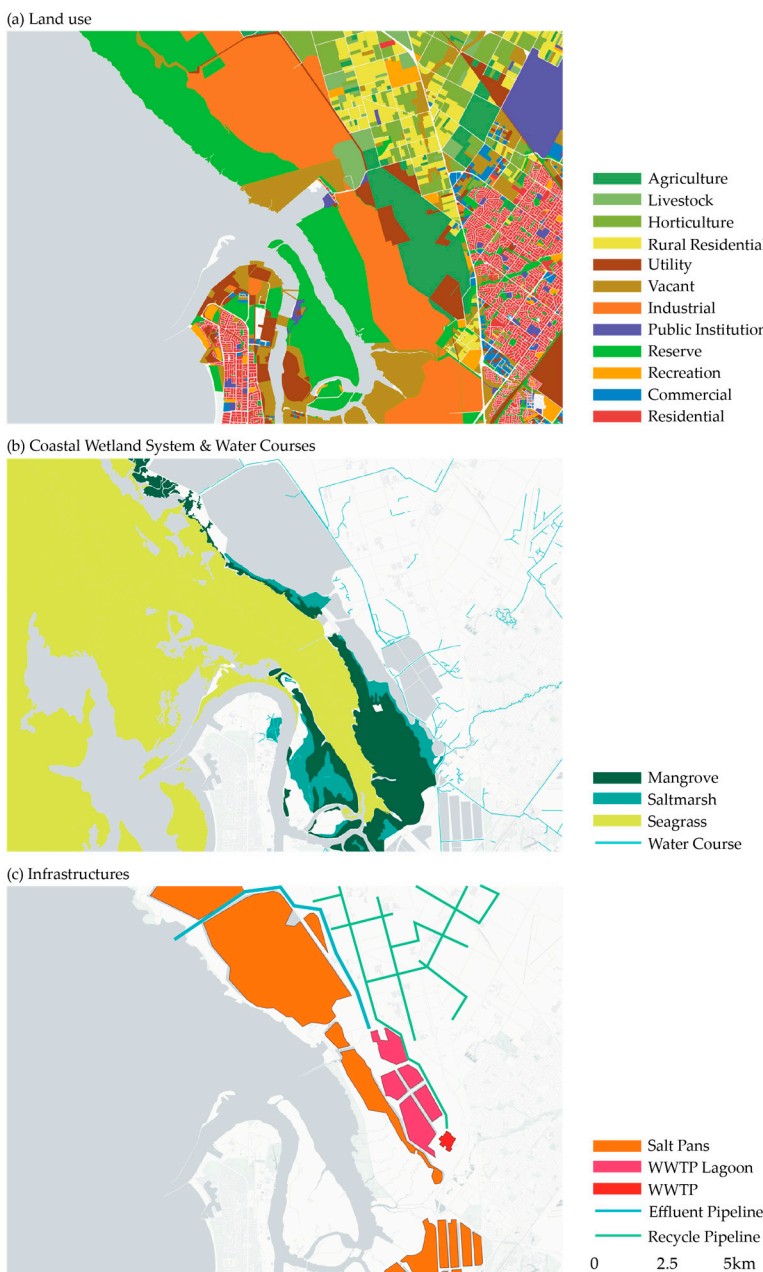

**Figure 6.** Mapping of urban systems and infrastructures surrounding the Bolivar WWTP. The interplay of these systems is central to each of the three developed scenarios. (Figure by authors using data from Geosciences Australia).

The Land Use Generalized layer is a regular parcel-based land use mapping data derived from evaluation and land division boundaries generated by Geoscience Australia in 2022. AEIP was created by Geoscience Australia and the Bushfire & Natural Hazard CRC in 2018, which aims to ensure the availability of nationally standardized exposure information. It includes various categories such as buildings, businesses, people, infrastructures assets and agricultural commodities throughout Australia. Through overlaying the various land use layers with simulated flooding maps for each scenario in a GIS using an overlay method, the area of each land use category submerged and the cost of reconstruction can be calculated. The per-km cost of the defensive wall construction was extracted from a recent defensive seawall construction in Australia and was used to calculate the defensive infrastructure works (see Table 3).

**Table 3.** The flooding area and the cost of infrastructure reconstruction and construction under each scenario. Note: Both the cost of infrastructure construction, reconstruction and the value of the coastal ecosystem in Tables 3 and 4 were inflation-adjusted to a 2022 price level.

| Data Details | | | Scenario 1: Business as Usual | | Scenario 2: Protection | | Scenario 3: Retreat | | Scenario 4: Accommodation | |
|---|---|---|---|---|---|---|---|---|---|---|
| *Data Reference* | *Classification* | *Detailed Land Use* | *Inundated Area (sqkm)* | | *Inundated Area (sqkm)* | | *Inundated Area (sqkm)* | | *Inundated Area (sqkm)* | |
| See AIEP [85] | Agricultural | Agriculture, Livestock, Horticulture | 6.05 | | 0 | | 6.05 | | 5.93 | |
| | Residential | Rural residential, Residential | 1.02 | | 0 | | 1.02 | | 0 | |
| | Commercial | Commercial, Recreation, Public institution | 1.32 | | 0 | | 1.32 | | 0 | |
| | Industrial | Industrial, Utilities, Industrial Vacant | 40.95 | | 0 | | 40.95 | | 37.39 | |
| | | **Total** | **49.34** | | **0** | | **49.34** | | **43.32** | |
| *Data Reference* | *Classification* | | *Number of Dwellings and Buildings to be Inundated* | *Reconstruction Value (million AUD)* | *Number of Dwellings and Buildings to be Inundated* | *Reconstruction Value (million AUD)* | *Number of Dwellings and Buildings to be Inundated* | *Reconstruction Value (million AUD)* | *Number of Dwellings and Buildings to be Inundated* | *Reconstruction Value (million AUD)* |
| See AIEP [85] | Agricultural | | - | 16.28 | - | 0 | - | 16.28 | - | 13.96 |
| | Residential | | 3711 | 940.40 | 0 | 0 | 3711 | 940.40 | 0 | 0 |
| | Commercial | | 102 | 8474.63 | 0 | 0 | 102 | 8474.63 | 5 | 1457.84 |
| | Industrial (Utilities are not included) | | 45 | 189.88 | 0 | 0 | 45 | 189.88 | 6 | 34.68 |
| | | **Total** | **3858** | **9621.19** | **0** | **0** | **3858** | **9621.19** | **11** | **1506.48** |
| *Data Reference* | *Classification* | | *Length of Sea Wall (km)* | *Cost (million AUD)* | *Length of Sea Wall (km)* | *Cost (million AUD)* | *Length of Sea Wall (km)* | *Cost (million AUD)* | *Length of Sea Wall (km)* | *Cost (million AUD)* |
| See McPhee [86] | Defensive Wall | | - | - | 23.93 | 487.71 | - | - | - | - |
| | | **Total** | **-** | **-** | **23.93** | **487.71** | **-** | **-** | **-** | **-** |

**Table 4.** The value of coastal ecosystem under each scenario in million AUD.

| Data Reference | Classification | Scenario 1: Business as Usual | | Scenario 2: Protection | | Scenario 3: Retreat | | Scenario 4: Accommodation | |
|---|---|---|---|---|---|---|---|---|---|
| | | *Area (sqkm)* | *Value (million AUD)* | *Area (sqkm)* | *Value (per million in 2005 AUD)* | *Area (sqkm)* | *Value (per million in 2005 AUD)* | *Area (sqkm)* | *Value (per million in 2005 AUD)* |
| See Kirkpatrick | Mangrove | 3.71 | 10.25 | 2.32 | 6.41 | 26.61 | 73.49 | 17.04 | 51.75 |
| [87] | Saltmarsh | 1.14 | 3.15 | 0.08 | 0.21 | 18.74 | 51.75 | 6.08 | 16.78 |
| | **Total** | **4.85** | **13.40** | **2.4** | **6.62** | **45.35** | **125.24** | **23.12** | **68.53** |

Additionally, under each scenario, the ecological value of the coastal ecosystem, including mangroves and coastal wetlands, was also calculated, and it is listed in Table 4. All of the costs were inflation-adjusted to 2022 price levels using an online inflation calculator hosted by the Reserve Bank of Australia [88].

After the schematic cost calculations for both infrastructure cost and ecological value were completed, a qualitative analysis was introduced to evaluate the future maintenance, ongoing repair cost and the risk embedded within each of the scenarios, and this is shown in Table 5. The expert evaluation was completed in the context of a landscape planning workshop in which future risk exposure and maintenance legacies were debated and scored using an ordinal scale of low, medium and high.

**Table 5.** Qualitative summary of the maintenance and ongoing repair cost and the risk for each of the four scenarios.

| | Scenario 1: Business as Usual | Scenario 2: Protection | Scenario 3: Retreat | Scenario 4: Accommodation |
|---|---|---|---|---|
| *Classification* | *Measurements* | | | |
| Maintenance and ongoing repair cost | High | High | Low | Medium |
| Risk | High | Medium | Low | Medium |

*4.4. Business-as-Usual Scenario (BAU)*

The business-as-usual scenario is an approach assuming that there will be no major human intervention to address the impacts of future SLR on the site. Therefore, it indicates an acceptance of current political inertia. It also provides a basis for a comparison with other approaches. As shown in Figure 7a, in this scenario, the whole on-site wastewater treatment system, including the WWTP and the lagoon, is under threat from storm surge, tidal flooding and seawater incursion.

Additionally, about 50 sqkm area of various land uses is submerged in the BAU scenario (Table 3). The cost of the reconstruction of the assets on these lands approximates AUD 9621 million. The existing bunds of the saltpans and wastewater treatment lagoon will also restrict the landward movement of mangroves and coastal wetlands, leading to the large-scale loss of ecological habitats. The contribution such natural assets provide to coastal resilience will also be lost as the habitats vanish. The mangroves and coastal wetlands play an important role in mitigating wave energy and reducing coastal erosion. With the disappearance of the coastal ecosystems, the risk and damage by future climate-related extreme weather events are likely to dramatically increase.

*4.5. Retreat Scenario 2*

Retreat scenarios reduce the potential exposure of coastal infrastructure to sea level rise and its induced hazards by moving infrastructure out of harm's way, either horizontally or vertically. There are different adaptation measures that can be used to reduce exposure risks, for example, raising the elevation of the existing facilities. In Boston, Massachusetts, the Deer Island Water Treatment Plant was raised about 60 cm to adapt to future sea level rise [89–91]. However, for some facilities, increasing their elevation is not feasible, and relocation and other strategies can be considered. For instance, a conceptual design of a wastewater treatment system in San Francisco separated the large stabilization lagoon into two parts. The first part consists of a constructed wetland that could be abandoned or converted into coastal wetlands at lower elevations in response to rising sea levels, with the second part sitting at a higher elevation to meet additional future needs and prepare for retreat [92].

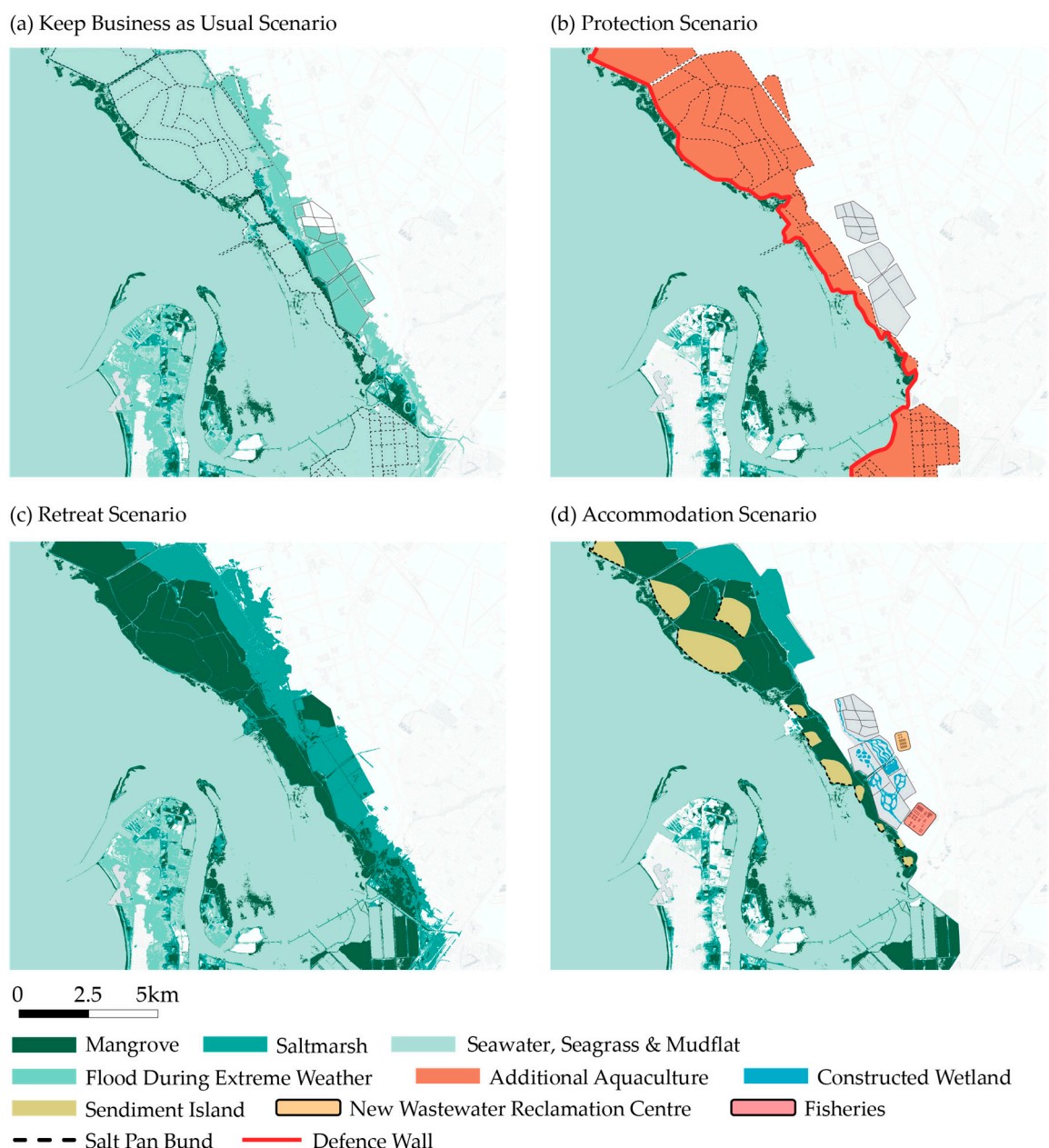

**Figure 7.** Mapping of urban and natural systems and infrastructure surrounding the WWTPs. The interplay of these systems is central to each of the three developed scenarios. The scenarios include the (**a**) business as usual scenario, (**b**) protection scenario, (**c**) retreat scenario and (**d**) accommodation scenario.

The Retreat scenario, as applied to the Bolivar WWTP site, uses the latest water reclamation technology developed in Singapore and proposes a new compact plant at an alternative higher elevation offsite to avoid future climate-related hazards. Lower vulnerable areas are abandoned, and existing levee banks are dismantled. By dismantling the levees, this strategy allows the coastal wetland system, including the seagrass meadow, mangroves and salt marsh, all of which rely on specific water depths to thrive, to progressively migrate inland as sea levels rise.

Therefore, the Retreat scenario (see Figure 7c) provides the maximum area of coastal wetland ecosystems in the future through facilitating inland ecosystem migration. It also provides a natural green infrastructure barrier for the new wastewater reclamation plant.

In this scenario, the coastal wetland system is assumed to have the capability to keep pace with the rising sea levels.

By partially dismantling the levees of the salt ponds and the stabilization lagoon of the Bolivar WWTP and modifying the terrain by extending existing water channels, the Retreat scenario allows the coastal wetland system to migrate and expand, creating an extensive wetland habitat. The expansion of the wetland and coastal ecological systems contributes to the biodiversity and habitat value of the metropolitan area. It also provides additional protection to the new wastewater reclamation plant against climate-related hazards. The coastal wetland system also reduces the nutrient loads of stormwater runoff and treated wastewater discharge, maintaining the health of the mangrove and seagrass and coastal ecosystems. Such ecosystems play a positive role in climate mitigation through blue carbon storage and support local and internationally valuable fish and wildlife populations [93]. Therefore, although this scenario performs strongly in terms of ecological outcomes, it also has broader economic benefits beyond the immediate conservation and tourism outcomes. A limitation of this scenario is uncertainty concerning the rate of sedimentation. According to Kirwan [94]), the sedimentation rate may increase at the same rate as the sea level rise or exceed that of sea level rise. The sedimentation rate may vary due to changing tidal patterns and sediment supply. Within the project area, the rate is uncertain [94].

*4.6. Protection Scenario 3*

Protection is an approach that involves using a range of physical and green infrastructure structures explicitly as defensive measures to safeguard against future sea level rise and climate-induced coastal hazards. Physical infrastructure, such as levees, dikes and seawalls, are examples of engineered structures that can defend against sea level rise. This approach can be applied at a variety of scales (country, city, community and even individual building). Most typically, engineered works are used to facilitate the protection of coastal areas and infrastructure facilities susceptible to rising sea levels and flooding.

Coastal settlements around Australia and globally have already experienced an increasing frequency of flooding and other impacts. To defend against coastal flooding, defensive structures, such as sea walls, embankments, levees and bunds, have been built. In Adelaide, most sections of the metropolitan coast are protected by rock walls, faced with boulders upon a foundation of smaller rocks to defend against the wave energy) [72]. Such artificial structures can reduce flooding impacts on the coastal area; however, this method affects the intertidal zone, leading to the loss of vulnerable ecosystem services and irreversible degradation of habitats. To somewhat ameliorate the impacts of grey infrastructure, green infrastructure, such as dunes and coastal wetlands, can also be constructed to reduce reliance on conventional engineered systems.

In this instance, the Protection scenario (see Figure 7b) involves the development of a seawall, re-using the existing onshore levees of the salt field and the coastal wetland system as a series of defensive structures. To mitigate the impact on ecosystems and habitat, a 'living' infrastructure strategy inspired by 'the living breakwaters' project in Boston [95,96] is applied, integrating and transforming the existing levees of the salt ponds. Instead of simply making the existing levees higher and stronger to defend against sea level rise, this strategy is applied to provide ideal habitats for a diversity of local species. A series of eco-concrete modules that mimic the reef habitat is used to reinforce the levees, allowing juvenile fish and shellfish to hide from predators underwater and hosting seals and nesting birds above water as well. The combination of the soft coastal wetland system and innovative living levees can attenuate the wave energy and protect the landward WWTP from flooding and other coastal hazards.

In addition, this scenario involves converting the currently abandoned salt ponds into high-value aquaculture fishery landscapes and coastal wetland reservation areas. Although this has a direct benefit for local economies, the living sea walls are designed to benefit the broader ecological landscape and wild fish populations too.

### 4.7. Accommodation Scenario 4 (Integration of Approaches)

Accommodation is an approach that integrates a variety of strategies and involves living with sea level rise by facilitating a more complex floodable landscape.

In this scenario (see Figure 7d), both built and emergent ecological systems are proposed. These include replacing the current wastewater treatment plant with a new compact wastewater treatment plant, such as that used at the Changi Water Reclamation Plant, Singapore [97], and reconstructing coastal ecosystems. The scenario converts the existing stabilization lagoon into a series of constructed wetlands and habitats. A hybrid coastal system is implemented that consists of natural coastal wetlands, constructed artificial wetlands, a more compact wastewater reclamation plant and high-value fisheries. In this scenario, efficient and safe wastewater treatment is provided through water treatment technologies and landscapes that provide a range of constructed landscapes and ecosystems that provide a range of ecosystem services [98], such as stormwater filtration, habitat restoration, and recreational and educational functions for the public.

The new landscape is conceived of as a novel ecosystem that is designed to be largely self-sustaining, resilient and adaptable to the rising sea level over time by enabling dynamic changes within the system rather than maintaining infrastructures to defend against them. Instead of reinforcing the existing levees in a defensive way, as in scenario 2, or dismantling them, as in the retreat scenario, this scenario cuts the levees into separated pieces, allowing coastal wetland systems to migrate inland. As mentioned in the Retreat scenario, the coastal wetland system has the capability to keep pace with the sea level through the vertical accumulation of sediments. The broken levees could facilitate this process by allowing trapped sediment to generate several new islands. Such islands could provide ideal habitats for both wetland vegetation and other estuarine and coastal fauna, as well as helping to attenuate wave energy.

Due to the increased efficiency of the proposed wastewater treatment and reclamation plant, the existing stabilization lagoon can be replaced by constructed wetlands, which provide habitat restoration, recreation and stormwater treatment functions. In this scenario, an innovative aquaculture farm is proposed for the current WWTP site using the recycled water from the wastewater reclamation plant, which then discharges the used water to the adjacent constructed wetland for purification. The constructed wetland is also able to transform into differing ecologies, such as mangroves and saltmarshes, as the system evolves in response to changing tidal patterns in a dynamic way. Therefore, there is no certain outcome in this scenario, but a number of possible pathways that may eventuate dependent on varying sea levels and resultant sedimentation patterns.

### 5. Concluding Remarks

This study aimed to investigate sea-level-rise-induced challenges to coastal infrastructure, specifically wastewater treatment plants and their associated landscapes. It identified both risks and opportunities for such WWTPs and developed creative approaches to adaptation through environmental and landscape design and planning. Adelaide was used as a case study as a city that shares similar challenges with many around Australia and the world. The challenges discussed in the study are typical and pressing and must be addressed carefully in order to build resilience and avoid disastrous outcomes for both people and ecosystems [11]. For this reason, scenario-based landscape planning was developed as an approach that broadens thinking and emphasizes creativity and alternative possibilities. The approach does not so much aim to identify ideal outcomes but rather presents a range of possibilities for further discussion and to raise awareness.

We have three main findings. Firstly, through creative scenario development focusing on the Bolivar Wastewater Treatment Plant, we find that whilst wastewater treatment plants are threatened by climate-related hazards, there is an opportunity for such systems and their associated landscapes to develop and be improved in response to the challenge of climate change. Secondly, with scenario-based design methods, we tested different ways to adapt existing infrastructures and land uses. The three strategies used include Retreat,

Protection, and Accommodation. These strategies each use different land use configurations to produce contrasting outcomes and provide choices and concepts for further discussion by communities and governments.

Finally, we demonstrate that there is a range of different ways of thinking about adaptive and resilient landscapes in uncertain economic and environmental contexts. Scenario-based approaches, as developed in this study, offer holistic and creative ways of dealing with large-scale infrastructure and coastal change and can inspire lateral thinking and provide choices in contexts that often succumb to prioritizing singular approaches without a comprehensive consideration of alternatives. In uncertain economic, environmental and social contexts, such approaches are significant, valuable and necessary in the way they broaden possibilities and expand thinking. As all of the developed scenarios contrast, they are useful in providing a range of topographic options that stimulate debate and can help dissolve destructive and adversarial political positions. Such scenario approaches are not templates to copy but rather ways of thinking that can help communities and governments consider the possibilities beyond business-as-usual approaches. Such visionary design methods support positive conversations and futures in a landscape commonly dominated by fear, denial and inertia. This research has taken place in an academic setting in a landscape architecture design studio with post-graduate students. The researchers can advance future exploratory scenario design research by directly engaging with the many urban governments and communities around Australia to assist them in developing their own unique and local approaches to resilience. Collaborative, community-focused research approaches that break down and personalize the monumental challenges of climate change are essential. Such activities can raise awareness of the potential of design and environmental planning to improve resilience in the context of major sea level rise in urban coastal environments.

**Author Contributions:** Conceptualization, K.Z. and S.H.; methodology, K.Z. and S.H.; software, K.Z. and S.H.; validation, S.H.; formal analysis, K.Z.; investigation, K.Z. and S.H.; resources, S.H.; data curation, K.Z. and S.H.; writing—original draft preparation, K.Z. and S.H.; writing—review and editing, S.H.; visualization, K.Z.; supervision, S.H.; project administration, S.H. All authors have read and agreed to the published version of the manuscript.

**Funding:** This research received no external funding.

**Data Availability Statement:** Not applicable.

**Acknowledgments:** The ideas for this paper were initially developed in the Landscape Architecture Program's LARCH7033 Final Design Studio at the University of Adelaide. The studio was led by Scott Hawken.

**Conflicts of Interest:** The authors declare no conflict of interest.

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
