# Peer review of "Climate-Related Sea Level Rise and Coastal Wastewater Treatment Infrastructure Futures: Landscape Planning Scenarios for Negotiating Risks and Opportunities in Australian Urban Areas"

_sustainability, doi:10.3390/su15118977_

Round 1

Reviewer 1 Report

This manuscript investigates the potential influence of rising sea levels on the wastewater treatment systems with the scenario-based landscape planning approach. I've enjoyed in reading this manuscript which is well-written and within the scope of Sust.. I'd like to recommend a minor revision before the publication of this manuscript. Below I detailed some comments to improve the readability of this manuscript.

1. I suggest the authors to summarize the main results of this study in the Abstract.

2. Lines 75-77, this paragraph can be deleted since the selection of the case study would be introduced in the following parts.

3. Lines 79-104, could the authors set a new paragraph to describe what the scenario-based approaches are?

4. Lines 110-113, please rephrase.

5. Lines 162-163, mangroves and vegetations.

6. Line 189, the approach of the study...

7. Line 204, what is the coastal squeeze, please clarify.

8. Lines 209-210, please rephrase.

9. Please simplify the conclusion and summarize the innovation of this study.

Reviewer 2 Report

Comments to Both Author and Editor:

Overall comments:

The authors address the effect of sea level rise on coastal infrastructure along the South Australian coastline, and in particular their effects on WWTPs.  This paper is timely in an environment where climate resilience is an important factor for protection and future development of coastal infrastructure.  Additionally, the effects on WWTPs in particular are not well documented.

The authors then present a several scenarios based on varying land use in Adelaide, (protection, retreat and accommodation) to compare their potential impacts.

While I agree with the approach of using various scenario modeling, I would have really preferred (and expected) some sort of introduction of certain quantitative comparisons (even just rough estimates) instead of discussing their impacts purely qualitatively.  The three scenarios appear to have wildly different costs to implement (capital and O&M). And what are the metrics by with these scenarios are comparable?  For example, it must cost a lot to protect the current coastline, but how much economic impact (gains and losses) would a complete retreat be for the economy of Adelaide?  It would be preferable to present some sort of cost benefit analysis arising from each scenario.

I recommend the paper to be published after a major revision, addressing this final point in particular.  Additionally, the paper needs to be proof-read in general as there is some cumbersome language.

Specific Comments on the text:

Line 70: Can the authors give a sense of the distance of these WTTPs from population centers?

Line 121: There is a switch between the use of WTTS and WTTPs.  If WTTPs are a type of WTTS are there other WTTS’s considered?  Or was this simply inconsistent use of terminology.

Line 213: please cite the data and methods used for SLR simulations.

Line 259-261:  What is an estimated percentage of Adelaide’s population that is served by the Bolivar WWTP (and could conceivably be affected by the scenario analyses)?

Figure 3 caption: check caption spelling

Figure 4 caption: grammar

Line 350: The Deer Island Water Treatment plant is in Boston, Massachusetts, not Virginia (I lived in Boston)

Line 352: “effective”, or “feasible”?

Lines 370-385:  The authors list many qualitative effects of the retreat scenario.  This really needs some sort of quantitative analysis, like how much will it cost to retreat and change the land use from urban to mangrove?  What are the economic impacts or relocating the population?

Line 387: How do we know that protection scenario 2 provides the maximum economic benefit?  There are no numbers or estimated cost comparisons in the paper.

Line 420: Again what are the costs?  At least provide an estimate.

Section 4.5:  Again, would prefer even a rudimentary quantitative cost comparison (even a rough estimate of $/km2) due to the switches in land use and the resultant losses/benefits

Reviewer 3 Report

The paper considers how scenario based landscape design approaches might enhance current debates and approaches to 17 coastal change with particular reference to wastewater treatment systems and associated environmental landscapes. Adelaide is used as the case study and a range of landscape-based simulations and scenario design approaches are developed and evaluated to assess the consequences and 20 potential of different courses of action including retreat protection and accommodation. The paper is very interesting. The following problems should be further observed:

(1)    There is no the organization of the paper in the Introduction section.

(2)    The figure.1 is not very clear. Please provide better figures with high resolution.

(3)     In Section 3. Methods and Approach, the presentation is a little simple. Please provide these methods in detail, such as equations, data, and how to process these data. Please pay attention to this point.

(4)    In subsection 3.5, For example, For each scenario we evaluated it in terms of social, ecological and economic out comes[38]. Please elaborate how do you evaluate the scenario.

(5)    In subsection 4.1, to select case study sites, three criteria were considered. Why do you consider these criteria?

(6)    In Section 4, three scenarios are discussed. Is there any other Scenario?

Round 2

Reviewer 2 Report

Thank you for the updated version of the manuscript.  I am very happy to see that the authors have taken many of my suggestions to heart and have made conscientious and significant efforts to present their much clearer methodology and quantitative results.  I think now the paper reads much better and sparks a lot more discussion and thought re: WWTPs and the various scenarios of land use that may be employed to protect a majority of Adelaide's residents, as well as to address habitat squeeze.  I therefore recommend publication of the updated version of the manuscript in its present form.

Author Response

Thank you for acknowledging our comprehensive revision and recommending publication.

Reviewer 3 Report

The paper has been revised well. Some future works can be further discussed at last. 

Author Response

Thank you to reviewer 3 for acknowledging our comprehensive revision and the improvements we have made to the paper based on their feedback.

We have proofed the paper again to correct some minor spelling errors and the like.

As reviewer 3 suggests we have also added a few sentences into the conclusion recommending future research